# Self-growing photonic composites with programmable colors and mechanical properties

Juan Xue[1,2], Xuewu Yin[1], Lulu Xue [3], Chenglin Zhang[1], Shihua Dong[1], Li Yang[1], Yuanlai Fang[1], Yong Li[1], Ling Li [4] & Jiaxi Cui [1,2] ✉

Many organisms produce stunning optical displays based on structural color instead of pigmentation. This structural or photonic color is achieved through the interaction of light with intricate micro-/nano-structures, which are "grown" from strong, sustainable biological materials such as chitin, keratin, and cellulose. In contrast, current synthetic structural colored materials are usually brittle, inert, and produced via energy-intensive processes, posing significant challenges to their practical uses. Inspired by the brilliantly colored peacock feathers which selectively grow keratin-based photonic structures with different photonic bandgaps, we develop a self-growing photonic composite system in which the photonic bandgaps and hence the coloration can be easily tuned. This is achieved via the selective growth of the polymer matrix with polymerizable compounds as feeding materials in a silica nanosphere-polymer composite system, thus effectively modulating the photonic bandgaps without compromising nanostructural order. Such strategy not only allows the material system to continuously vary its colors and patterns in an on-demand manner, but also endows it with many appealing properties, including flexibility, toughness, self-healing ability, and reshaping capability. As this innovative self-growing method is simple, inexpensive, versatile, and scalable, we foresee its significant potential in meeting many emerging requirements for various applications of structural color materials.

Structural coloration, which was first observed by Robert Hooke and Isaac Newton on peacock's feathers more than 350 years ago, constitutes a class of fundamentally different mechanisms to generate color in contrast to chemical or pigment-based coloration[1–3]. Here the colors are produced by the microscopical structures or surfaces fine enough to interfere with light and thus show appealing features such as brilliant, stable, and non-toxic[4]. This phenomenon has initiated the field of artificial structural color materials and inspired many applications in optoelectronics[5], anti-counterfeiting[6], sensors[7], etc. Essential

to these applications is the fabrication of various sophisticated periodic photonic nanostructures[8–11]. To this end, both top-down and bottom-up technologies have been developed[12,13]. The former mainly depend on microfabrication processes[14] that often suffer from high processing cost and low production efficiency, which hinders their scalable manufacture of structural color products. In contrast, bottom-up methods in which ordered nanostructures are made through the self-assembly of nano-/microscopic building blocks are simple, cheap, and scalable[15]. In self-assembly methods, building blocks with uniform

[1]Institute of Fundamental and Frontier Sciences, University of Electronic Science and Technology of China, No. 5, Section 2, North Jianshe Road, Chengdu, Sichuan 610057, P. R. China. [2]Yangtze Delta Region Institute (Huzhou), University of Electronic Science and Technology of China, Huzhou 313001, P. R. China. [3]Department of Bioengineering, University of Pennsylvania, Philadelphia 19104, USA. [4]Department of Mechanical Engineering, Virginia Polytechnic Institute and State University, 635 Prices Fork Rd, Blacksburg, VA 24060, USA. ✉e-mail: Jiaxi.Cui@uestc.edu.cn

sizes are required to achieve ordered nanostructures, which leads to single or pseudo photonic bandgap and hence monochrome or iridescence. For many applications, multicolor patterns are necessary, such as cosmetics, apparel, security, bioanalysis, sensors, electrical appliances, and automobiles[16]. Patterning structural-color materials can significantly enhance their added value. To achieve multicolor images, several methods have been developed recently, including confinement deposition/swelling[17], regioselective removal[18], post-modification of inverse opal[19], etc. Among them, selective swelling[20,21] can even bring about interesting stimuli-responsiveness and reversibility. For swelling-induced patterning, region-dependent swellability is required, and the liquid-containing nature might also induce instability. By contrast, photopolymerization allows direct patterning by immobilizing compounds to modulate the bandgap[22], but the variation in bandgap is limited due to the restriction of curing rigid substrates. Generally speaking, these methods typically involve elaborate fabrication techniques requiring superb synthetic skills, yet the obtained images are either ephemeral, low-resolution, or relatively dull, stiff, and low-controllable. Moreover, artificial structural color materials are often fragile and not reprocessable[23,24], in stark contrast to their glorious, durable, and lightweight natural counterparts. Therefore, it is highly desirable to develop facile approaches for fabricating bright, robust, and adaptive structure color patterns.

The intricate microstructures for structural colors of living creatures are produced via a natural growing process[25–28]. Take the brilliantly colored eye pattern of the peacock's tail as an example (Fig. 1a). The barbules within this coloration pattern contain two-dimensional (2D) photonic crystals made up of melanin rods and air holes embedded in a keratin matrix[29]. By controlling the growth of keratin during feather development, peacocks can vary the spacing between adjacent melanin rods, which effectively adjust the lattice constant to modulate the photonic bandgaps for producing diversified colors (Fig. 1b and Supplementary Fig. 1). Such a sustainable approach not only allows living organisms to continuously and precisely evolve their colors and patterns on original coats, rather than to generate new substrates with a set of components with different sizes, but also mechanically robust matrices, implying a fundamentally innovative, efficient, sustainable method to synthesize or mediate structural color figures.

Mimicking the growth of living organisms with synthetic building blocks is an emerging strategy for designing intelligent materials[30,31]. We have recently developed an original approach to enabling cross-linked polymers to incorporate externally-provided compounds to grow[32]. The growth involves swelling and polymerization of precursors as well as homogenization of the newborn and original networks. Such design allows us to create structural surfaces from smooth substrates[33], endow rigid substrates with self-healing ability[34], and fabricate conductive hydrogel circuits[35]. However, the growing systems currently available all possess simple polymer matrices, far away from the fine, ordered, complex composite structures observed in peacock's tail feathers. Herein, we propose to apply the growth concept to fabricate photonic composites with controlled structural coloration, patterning capability, and versatile mechanical properties. As shown in Fig. 1c, this material system is composed of SiO$_2$ nanospheres embedded in a "living" polymer matrix, which can undergo spatial-selective growth by introducing "nutrients" in the external environment. The growth-induced expansion of the lattice parameters allows the system to precisely and continuously vary its photonic bandgaps and thus color in an on-demand manner. More specifically, the growth is achieved via homogeneously swelling the crosslinked polymer matrices using a nutrient solution consisting of monomer, crosslinker, photoinitiator, and catalyst simultaneously, followed by light-induced polymerization. A new-old double network structure forms, in which the original network is stretched. Note that the stretched conformation would restrict the uptake of more nutrient solution for further

growth. The catalyst in the nutrient solution is thus designed to remove such restriction by inducing chain exchange between new and old networks to homogenize the matrices, allowing the system to grow further (see Supplementary Fig. 2 for the chemical reactions designed for the growth). With this design, we can continuously change the photonic bandgap and system compositions for modulating materials' colors and mechanical properties on demand. Due to the spatio-temporal advantages of light, it is facile to generate multicolor images/patterns.

## Results
### System and synthesis

Acrylate-based polymers were selected as the growing matrices because of their widespread usage in various applications. The initial photonic composite films were made from SiO$_2$ nanospheres (Supplementary Figs. 3 and 4) and poly(ethylene glycol) diacrylate (PEGDA). Briefly, the suspending solutions of SiO$_2$ nanoparticles and PEGDA in ethanol were allowed to evaporate to induce the self-assembly of SiO$_2$ into photonic composites, followed by curing under UV irradiation (Supplementary Fig. 5) in homemade containers. After annealing, vivid films were obtained. The formation mechanism and driving force of SiO$_2$ forming an ordered structure were polymerization-induced colloidal assembly, which had been carried out by Ge et al.[36]. The average thickness of these vivid films is 0.14 mm. We found that changing film thickness (in the range of 0.1–0.2 mm) made no contribution to film swellability and color (Supplementary Figs. 6, 7). Thus, the films with a thickness of 0.14 mm were employed for further study. Scanning electron microscopy (SEM) investigation illustrates that the samples consisted of long-range ordered and short-range ordered domains (Supplementary Fig. 8). The long-range ordered structures would lead to iridescence while the short-range ordered one would result in angle-independent colors[37]. In our samples, long-range ordered domains were minorities. They were dispersed randomly in short-range ordered matrices, forming amorphous structures. As a result, the colors are angle-independent (Supplementary Fig. 9). The wavelength ($\lambda$) of the reflected light to this class of structural color films could be estimated by the Bragg's diffraction equation #1[38]:

$$m\lambda = 2dn_{eff} \tag{1}$$

where $d$ refers to the average interparticle distance, $n_{eff}$ indicates the mean refractive index of the composites, and $m$ is the order of reflection ($m = 1, 2,...$). In our design, $d$ could be modified by allowing the polymer matrices to grow.

The mixture of an acrylate monomer, 1,6-hexanediol diacrylate (HDDA, crosslinker), 2-hydroxy-2-methylpropiophenone (photoinitiator), and benzenesulfonic acid (BZSA, transesterification catalyst) was used as the nutrient solution. Here, three kinds of monomers were employed, including 4-hydroxybutyl acrylate (HBA), PEGDA, and 2-hydroxyethyl methacrylate (HEMA). HBA is a commonly used elastomer precursor, while PEGDA is the compound used to prepare the initiated sample and therefore would not vary the composition of the sample matrices during growth; compared to HBA and PEGDA, HEMA could form rigid polymer main chains and thus provided an approach to tune material mechanical properties. The nutrient solution containing HBA, PEGDA, or HEMA used for growth was defined as nutrient solution $B$, $EG$, or $M$, respectively. For growth, as-prepared purple films (typical mass percentage of the polymer matrix: 40 wt%) were selected as the initial samples. They were first immersed in a nutrient solution for swelling. During swelling, the samples increase in size and change their colors (Fig. 2a). The samples could swell the tested nutrient solutions with equilibrium swelling ratios of 9.1 wt% ($B$), 6.2 wt% ($EG$), and 7.1 wt% ($M$), respectively (Supplementary Fig. 10). The relatively low swelling ratios were attributed to the low polymer fraction in the

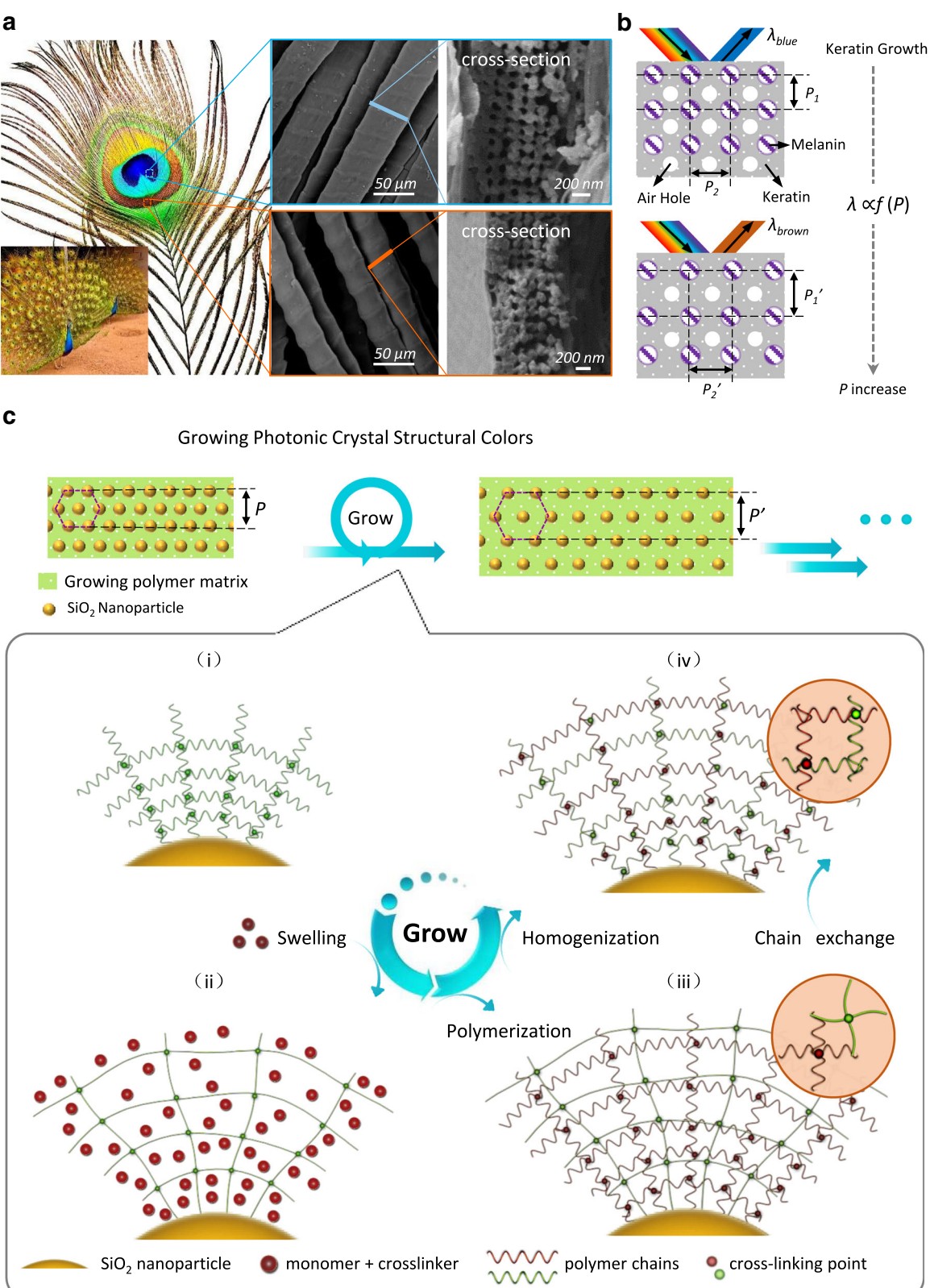

**Fig. 1 | Growing photonic composite structural color. a** Images of a peacock and its tail feather. The SEM images show the blue (up) and brown (down) barbules and their transverse cross-section structures. **b** Schematic diagrams of the photonic composite structure of the structural colors generated by a peacock. Selective keratin growth leads to different structural colors and patterns. **c** Schematic diagram of the growth of a SiO₂/polymer photonic composite: the polymer matrix with a crosslinked structure (i) is swelled by a solution containing polymerizable compounds to change its size (ii); after polymerization, a new-old double network structure (iii) forms, in which the old network (green line) is stretched; a chain exchange-based homogenization process is triggered to get a uniform matrix with relaxed chain conformation (iv) which allows for further growing cycles.

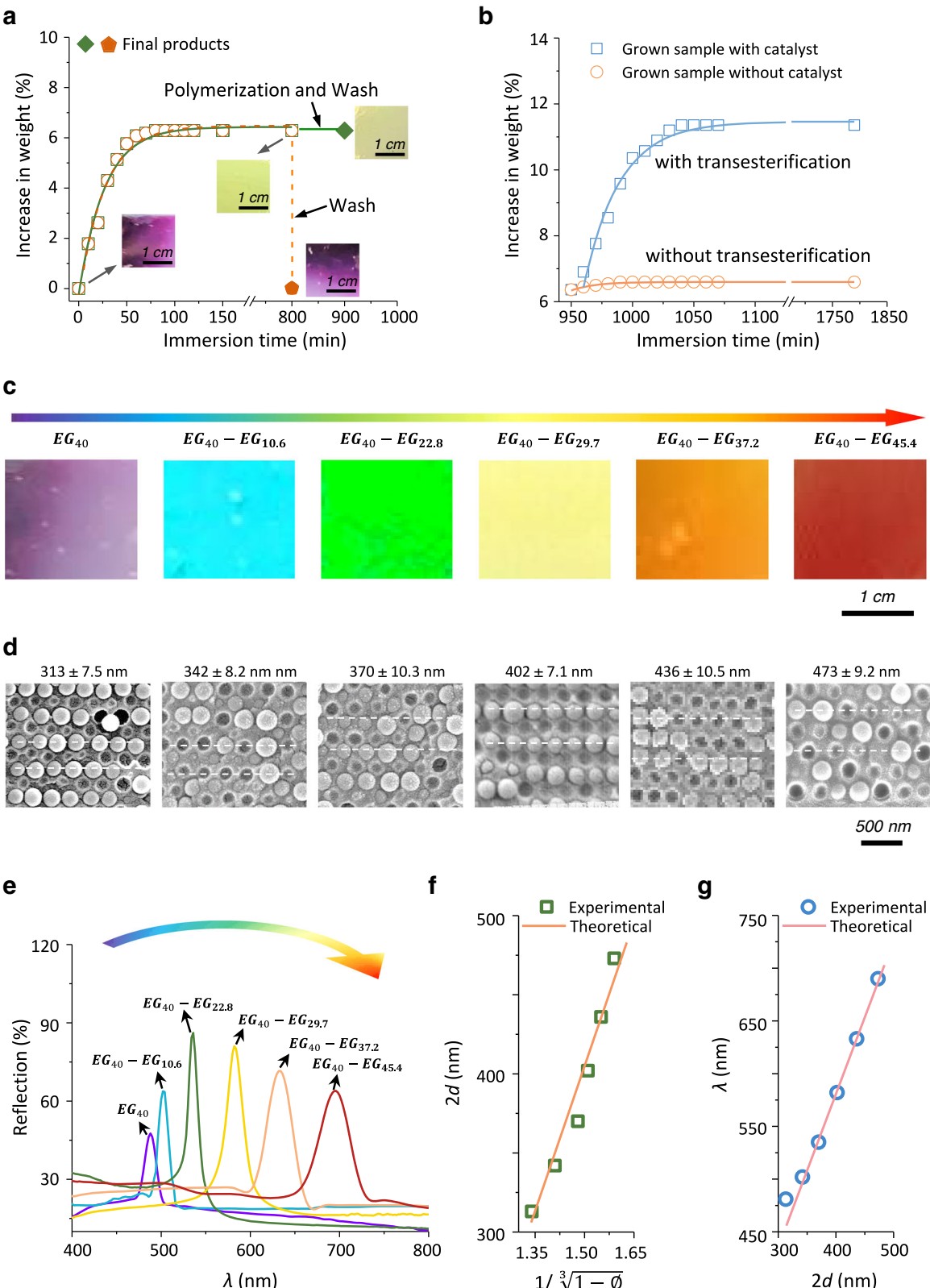

**Fig. 2 | Optical properties of grown photonic composite films. a** Time-dependent weight of $EG_{40}$ immersed in nutrient $EG$ (the mixture of monomer PEGDA, crosslinker, photoinitiator, and transesterification catalyst). Inserts are the digital photos of samples at different states. Ethanol/CHCl$_3$ solution was used for washing free nutrients. **b** Time-dependent weights of different grown samples immersed in a new nutrient $EG$. The catalyst-containing/free grown samples were obtained from nutrient $EG$ with/without transesterification catalyst. Digital electronic photographs (**c**), cross-section SEM images (**d**), and UV-Visible spectra (**e**) of $EG_{40}$ and its grown products. **f** Experimental and theoretical interplanar spacing $2d$ versus $1/\sqrt[3]{1-\varnothing}$ of $EG_{40}$ and its grown products, where $\varnothing$ is the volume fraction of the polymer matrix. The solid line is the theoretic curve. **g** The theoretical and experimental reflection wavelength $\lambda$ versus interplanar spacing $2d$ of $EG_{40}$ and its grown products. The solid line represents a theoretical line obtained via equation #**1**, where a composite $n_{eff}$ of 1.453 (20°C) is used.

photonic composite films and their rigid nature. Despite low swelling ratios, obvious color changes were observed. The colors were unstable and would turn back to purple again if the samples were washed to remove the entrapped nutrient solution. Besides, the swelling significantly weakened the Young's modulus of the samples (Supplementary Fig. 11). By contrast, the samples that were subsequently irradiated by UV light (365 nm, 10 mW·cm$^{-2}$) to trigger photopolymerization preserved their after-swelling weights and colors, as well as became tougher (Fig. 2a & Supplementary Fig.e 12). Three kinds of UV lights with different intensities (5, 10, and 20 mW·cm$^{-2}$) have been applied to induce the polymerization. The obtained samples show nearly identical spectra (Supplementary Fig. 13), suggesting that the UV light intensities had a negligible impact on the displayed colors once the adopted light was strong enough to induce homogeneous polymerization. The photopolymerization reaction was exothermic, which could heat the systems to 47.1ºC in 50 s (Supplementary Fig. 14). Such thermal effects could trigger transesterification reactions to release any polymerization-induced mechanical tension in such dynamic networks. As a result of such relaxation, the grown samples could swell nutrients again, showing similar swelling ratios with the original samples (Fig. 2b). This feature allowed repeated growth. To illustrate the role of this homogeneous step in growth, a control sample without transesterification catalyst was treated under the same process aforementioned. As expected, this catalyst-free sample was nearly non-swellable to the nutrient solution after the first growing cycle. In principle, the growing cycles could be repeated indefinitely to continuously red-shift the color, even though we only focused on the visible range in the current study. We denoted the initial sample as $EG_n$ where $n$ is the mass percentage of the polymer matrix, while the grown samples as $EG_n - B/EG/M_m$, where $m$ is the increased mass of corresponding polymer matrices compared to the entire original sample. For example, specimen $EG_{40} - B_{9.7}$ was the grown sample of $EG_{40}$ obtained from nutrient $B$ with a net increase of 9.7 wt% compared to initiated weight (total polymer fraction: $(40 + 9.7)/(100 + 9.7)$ wt%). All the specimens, including both initiated and grown ones, were self-standing and macroscopically uniform.

## Optical property

The homogeneous growth allowed for flexible modulation of sample colors in the full visible light spectrum from purple to red (Fig. 2c). During growth, SiO$_2$ colloidal nanoparticles maintained their ordered arrangement in the polymer matrix but showed increasing interplanar lattices, as evidenced by the gradual change of the double interplanar spacing $2d$ from 313 to 473 nm collected from the SEM images (Fig. 2d). UV-Visible reflection spectroscopy was employed to characterize the optical properties of different grown samples (Fig. 2e). With the increase in polymer matrices (then lattice distances), the reflection peak ($\lambda$) shifts from 481 to 690 nm. These reflection peaks were surprisingly intensive and narrow, implying a narrow distribution of SiO$_2$ colloid bandgap. The reflection peaks became slightly broader with growth, implying a decrease in the ordered degree of the SiO$_2$ colloid glasses. We attributed this decrease to the growth-induced magnification of the inhomogeneity of the short-range ordered structures that induced the structural colors. Compared to the grown samples, the samples with similar composition could also be directly prepared via one-step polymerization. However, we found that the grown samples displayed sharper reflection peaks than those obtained from one-step polymerization (Supplementary Fig. 15), implying the advantage of our growth strategy in tuning the colors. In our self-growing photonic composites, the changes in both double interplanar spacing $2d$ and reflection peaks (the colors) were predictable. For a photonic composite with a homogeneous ordered structure, the double interplanar spacing $2d$ was proportional to the cube root of the volume fraction of SiO$_2$ nanoparticles ($1/\sqrt[3]{1 - \varnothing}$, where $\varnothing$ is the volume fraction of the polymer matrix). By using the experimental data obtained from the

initial photonic composite to establish the linear relationship (See section **7** in supplementary information for calculation detail, Supplementary Table 1), we found that the theoretic curve compared well with the experimental $2d$ values (Fig. 2f), suggesting that the $2d$ could be precisely controlled by growth. Furthermore, in our system, the polymer showed a similar refractive index $n$ with that of SiO$_2$ (SiO$_2$: 1.45; polymer: 1.47 at 20ºC), and therefore, the $n_{eff}$ could be considered as a constant during growth. Note that the small refractive index contrast would lead to weak reflection. In our case, the samples were quite thick so that strong multiple scattering would occur to induce brilliant structural colors[39]. It indicated a linear relationship between $\lambda$ and $2d$ based on equation #1. When we plotted the experimental $\lambda$ to $2d$, they all fit well with the theoretical curve (obtained from equation #1 with a $n_{eff}$ of 1.453, Fig. 2g). All these results indicated that the optical properties of photonic composite films could be precisely post-modulated by growth.

## Mechanical properties

Growth allowed the samples to vary their compositions for modulating their mechanical properties. Two kinds of approaches could be applied, including changing the monomer types or the crosslinker concentration in the nutrient solution. For example, in the case of using crosslinker-free nutrient solutions, the sample would maintain its modulus when grown from nutrient $EG$ but become softer or stiffer from nutrient $B$ or $M$, respectively (Fig. 3a). On the other hand, increasing crosslinker concentration in the nutrient solution stiffens the samples regardless the monomer types (Fig. 3b).

Besides stiffness modulation, growth also significantly toughened the photonic composite structure. The as-prepared initial samples were brittle and would fracture upon bending (Fig. 3c). Such brittle samples could grow to be fully flexible by adding agents that could soften the matrices in the nutrient. In the current system, the composites were stiffened by both the highly crosslinking structure of polymer matrices and the covalent connection between SiO$_2$ nanoparticles and polymer chains of matrices (Supplementary Figure 16 & 17, FTIR results indicated the existing of the transesterification between the ester polymer and the hydroxyl on the surface of SiO$_2$ nanoparticles during the preparation of the initiated samples). Since both linkages were ester-based bonding, alcohol that could hydrolyze the ester connection was selected as the additive (Fig. 3d). As expected, the grown sample obtained from an alcohol-contained nutrient was very flexible. More than bending, the flexible grown sample can be rolled up, twisted, or folded (Fig. 3e) and shows fair elasticity. The sample became more stretchable without a tradeoff in strength (Fig. 3f). Here, the alcohol could either be used as an additive in normal nutrient solutions or an independent agent (a nutrient consisting of only alcohol) to make the sample flexible (Supplementary Fig. 18). The transesterification-based softening mechanism was confirmed by using a SiO$_2$-free PEGDA film in which the contribution of SiO$_2$ nanoparticles to material mechanical properties was absent. The film was grown in alcohol-free and alcohol-containing $EG$, respectively. The grown sample made from an alcohol-containing nutrient shows a nearly 50% increase in flexibility compared to the grown sample prepared by alcohol-free $EG$ (Supplementary Fig. 19). To further prove the mechanism, hydroxyl-free agents like tetrahydrofuran were employed as the additive, and none of them could make the sample flexible (Supplementary Fig. 20). Traditional photonic composite-based structural color materials are normally brittle and improving their mechanical properties thus constitutes one of the most critical aspects of current research[40,41]. Two strategies, i.e. softening photonic composite materials by the use of flexibilizers or increasing polymer fractions[42] and fabricating full organic-based inverse opal structures[43], had been developed. In the former, increasing polymer/flexibililizer fraction would disturb the self-assembly of particles and thus destroy the ordered colloidal microstructure. The materials with inverse opal

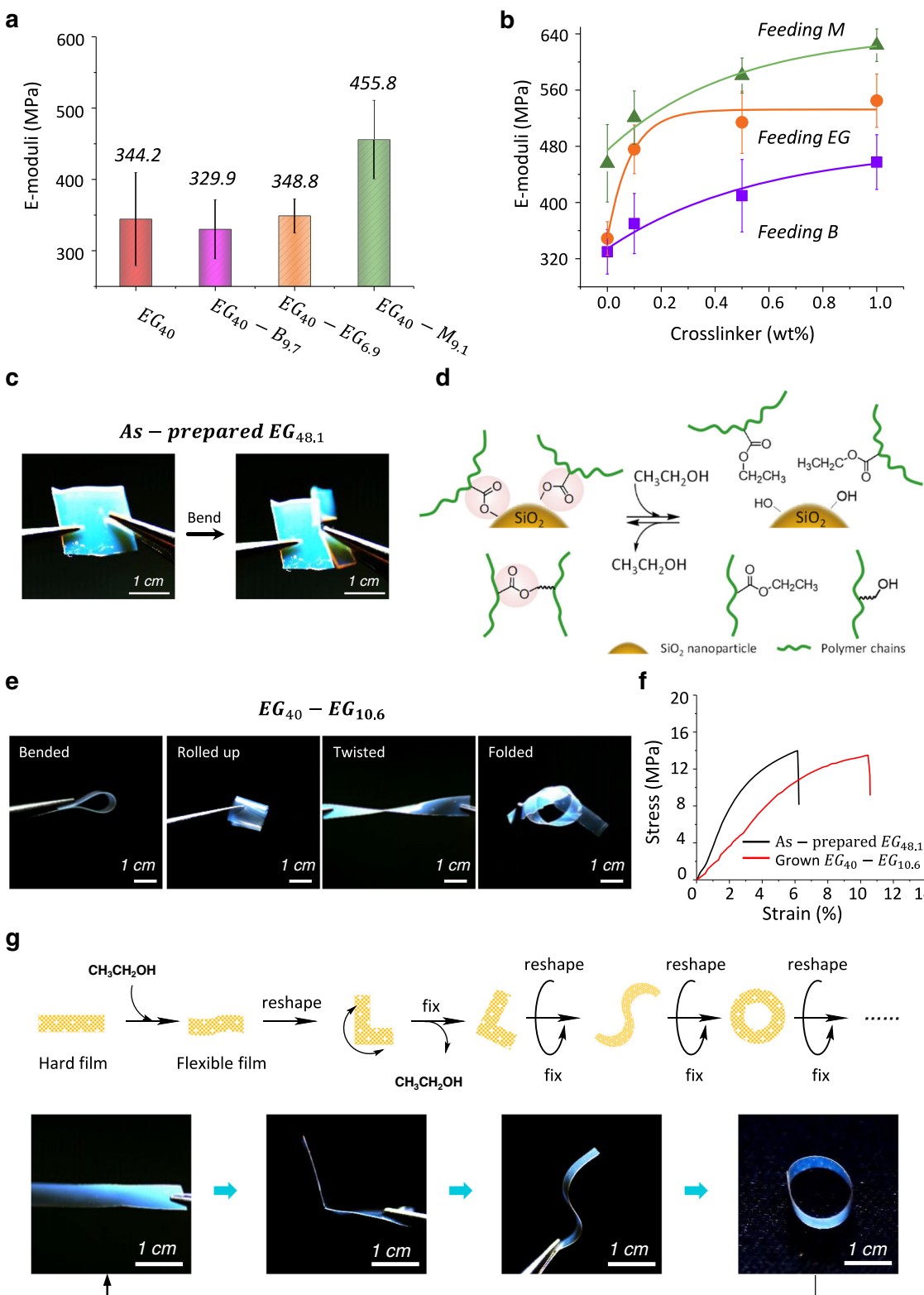

**Fig. 3 | Mechanical property of grown photonic composite films. a** Elasticity moduli of $EG_{40}$ and its grown samples obtained from different crosslinker-free nutrients. **b** Elasticity moduli of $EG_{40} - B_{9.7}$, $EG_{40} - EG_{6.9}$ and $EG_{40} - M_{9.1}$ samples obtained from the nutrients with different crosslinker concentrations. The E-moduli in **a** and **b** were obtained from five independent measurements. **c** Digital photos of the fracture of an as-prepared $EG_{48.1}$ sample during bending. **d** Chemical mechanism of reversible softening: alcoholysis to dissociate the connecting structure (softening) and removal of the alcohol moieties to reform the connections (stiffening). **e** Digital photos of a flexible $EG_{40} - EG_{10.6}$ sample (final $EG$ fraction: 45.8%). **f** The strain-stress curves of $EG_{48.1}$ (as-prepared) and $EG_{40} - EG_{10.6}$ (grown). The samples have similar compositions but the grown one shows a larger strain. **g** Reshaping mechanism (above) and the digital photos of a multiple-time reshaping sample. For reshaping, the sample was first softened by uptake of alcohol and deformed into a designed shape, followed by annealing treatment to remove the alcohol to fix the desired shape.

structures show excellent flexibility, but their preparation normally required HF-based etching treatment and the materials display milky appearance and low color saturation due to the higher light scattering in inverse opal[44]. In contrast, the growth method allowed for post-modulation of the composition without compromising the ordered microstructure, implying a more efficient method to get mechanically robust photonic composite materials.

## Reshape

The alcohol-based transesterification reactions were reversible and the broken ester linkages could reform by removal of the absorbed alcohol (Fig. 3d). To elucidate this idea, the softened PEGDA film made from the alcohol-containing nutrient was annealed at 70°C for 4 h to remove the alcohol moieties. The grown sample became stiffer and showed a modulus similar to that of the grown sample directly obtained from the alcohol-free nutrient (Supplementary Fig. 19). Such reversible softening-stiffening process allowed for multiple-time reshaping of the samples. As shown in Fig. 3g, a rigid flat film was first softened by growth with alcohol and then deformed, followed by annealing treatment (to remove the alcohol moieties) to fix the obtained structures. The reshaping could be repeated to achieve different forms. Interestingly, the deformed sample maintained its new shape when it was turned into an elastic state. Compared to this processability, traditional structural color materials are either rigid (photonic composite, would fracture) or elastic (inverse opal material, turn back to original shape), and do not show such reshaping capability.

## Spatial-selective growth and multicolor patterning

Another significant advantage of this photonic composite system is its spatially selective growth capability by using patterned photoirradiation (Fig. 4a, see Supplementary Fig. 21 for the detailed patterning procedure of selective growth). As shown by the surface profile, the irradiated region grew up with a sharp boundary. A height drop of 9.5 μm (yellow-orange change) occurred in ~1 μm distance (aspect ratio: 9.5). Consequently, the grown region showed a significantly different structural color from the unirradiated region. Note that the modulation could be achieved without the removal of the unreacted nutrient solution. In a previous study[33], we have proved that region-selective conversion of the monomer and crosslinker into polymer matrices by photopolymerization would induce a concentration gradient of the monomer and crosslinker. Such gradient would drive the nutrient to transport from unirradiated region to irradiated one, leading to the formation of convex surface textures at the irradiated region. The same mechanism was expected here. After growth, the residual nutrient entrapped in the composite was still active and could be transported to new irradiated regions for creating other surface textures and colors. The structure of the grown region was further probed by SEM. In the cross-section image of a grown sample, a sharp boundary was also observed. In the grown region, $SiO_2$ nanoparticles distribute orderly as those in the unirradiated region but show a larger interplanar lattice, implying homogeneous expansion of the polymer matrices.

Localized growth of photonic composite films indicated an innovative approach for chromatic patterning. We fabricated a "Sichuan facebook" using a $EG_{48}$ substrate with single blue color to demonstrate the flexibility of this approach (Fig. 4b). The blue substrate became olivine after swelling in $EG$. When such swollen substrate was subjected to UV light under a mask, yellow "Sichuan facebook" profile was outlined. After the unreactive nutrients were removed, the irradiated regions maintained their yellow color while the unirradiated parts turned back to light blue. The substrate was then re-swollen with $EG$ for further selective irradiations twice through different masks (Supplementary Fig. 22), leading to the formation of new structural colors and a clear multicolor image. This growth-based patterning approach differed from the reported methods based on either complicated instruments[45,46] or sophisticated skills. Here the structural color was patterned by user-friendly photo-lithography to directly achieve complicated, high-resolution images. The obtained pictures were not only stable but also always permitted further modification, due to the "living" growing mechanism, implying a more facile and effective direction for batch manufactures.

## Self-healing

The self-healing of mechanical damages on structure colored materials was a big challenging because of their rigid nature[47,48]. Since growth allowed for selective formation of new matrices in the irradiated region, it opened up a new approach for our composite photonic system to restore the damage. To demonstrate this self-healing capability, a photonic composite structural color film was first scratched on its surface. Although the polymer matrix itself is self-healable at high temperatures due to the transesterification-based chain exchange reaction[49], the scratch did not effectively repair upon heating treatment (Supplementary Fig. 23). In contrast, when we selectively induced growth on the damaged region by light irradiation, the scratch was completely repaired, and an intact photonic composite film was achieved (Fig. 4c). The healed film maintained its vivid structural color through the whole body (color in the healed region might slightly shift due to the change in the photonic bandgap, Supplementary Fig. 24). This growth-based self-healing method differed from traditional intrinsic and extrinsic self-healing strategies. In the intrinsic self-healing, no new matrix would form and the polymer chains should be mobile enough to undergo reconfiguration[50,51]. It is difficult for a rigid substrate to self-heal through this mechanism. Extrinsic self-healing strategy allows for the formation of new matrices but no ordered structure forms in the healed matrices[52]. Moreover, an interface between original and healed matrices often appears, which is unfavorable for the recovery of the mechanical properties. Here, our method combined the merits of both intrinsic and extrinsic strategies, and the healed sample completely restored its color and mechanical properties. The healed region exhibits comparable mechanical strength as the original sample (the self-healing efficiency: 80.1%, evaluated by the work, Fig. 4d), therefore, when stretched, the sample may even break at the intact region, rather than the healed one. By contrast, the scratched sample without self-healing was easily broken (8.8%).

## Discussion

Inspired by the forming process of peacock's tail feathers, we have proposed a fundamentally innovative strategy for fabricating photonic structural color composites. This strategy involves the uniform expanding of the polymer matrices of the structural color composites by allowing the matrices to grow, in which the finely ordered microstructure is maintained. By varying the component in the nutrient used for the growth and the growing cycles, the samples can be selectively changed their sizes, color, and mechanical properties. Under this mechanism, the obtained materials not only show good flexibility, toughness, and self-healing/reshaping ability but also permit the fabrication of complicated, high-resolution patterns. Since this facile method can be applied to commonly used polymer systems and results in excellent performances, we foresee its great application potential in meeting many emerging challenges of structural color materials and broadening their commercial values.

## Methods

### Materials

4-Hydroxybutyl acrylate (HBA, refractive index $n$ (20°C): 1.452), poly(ethylene glycol) diacrylate (PEGDA) (average Mn = 250 g·mol$^{-1}$, $n$ (20°C): 1.457), ammonia (28 vol%), 2-hydroxy-2-methylpropiophenone (photoinitiator), 1,6-hexanediol diacrylate (HDDA), and benzenesulfonic acid (BZSA, transesterification catalyst) were purchased from Aladdin Co. (Shanghai, China). Tetraethylorthosilicate (TEOS, 99.5%),

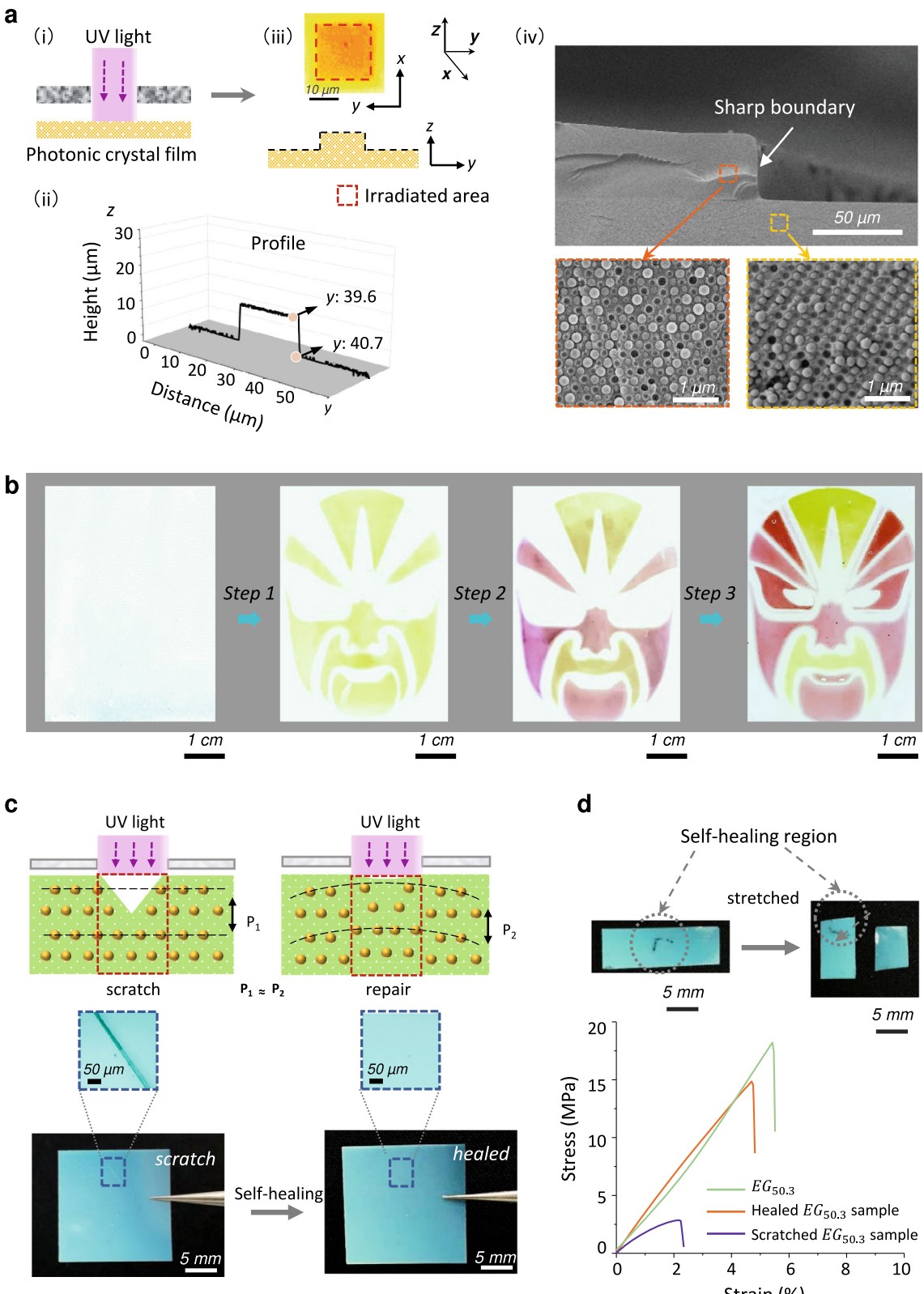

2-hydroxyethyl methacrylate (HEMA, 99%, *n* (20°C): 1.441), tetra-hydrofuran (THF, 99.8%, anhydrous), and ethanol (AR, ≥ 99.8%) were purchased from J&K (Beijing, China) and Aladdin Reagent Co. (Shanghai, China). Ethyl decanoate (98%) was brought from Tansoole (Beijing, China). All reagents and solvents were used without further purification. Ultrapure water (18.25 MΩ·cm, 25 °C) was used. The peacock feathers were purchased from a Taobao store (OPAI diy) and

photographed by the author. The image of the peacock were taken by the author at the farm in Chengdu.

## Synthesis of SiO₂ nanoparticles

$SiO_2$ nanoparticles were prepared via a modified *Stöber* method involving the hydrolysis and condensation of TEOS in ethanol in the presence of ammonia (catalyst)[53]. Briefly, the mixture of 100 mL ethanol and

**Fig. 4 | Spatial-selective growth of photonic composite films. a** Photo-induced selective growth. i. Schematic diagram of the light-induced growth process; ii. The profile of the grown sample; iii. Top view of a grown sample obtained from light-induced growth using a square photomask with $20 \times 20 \, \mu m^2$; iv. Transverse-section SEM image (top) of a grown sample and zoom in on the red and orange areas to obtain high magnification SEM images (bottom). **b** Fabrication of multicolor "Sichuan facebook" from a substrate with single blue color. $EG_{48}$ was used as the starting substrate and the nutrient $EG$ was used for growth. The specific preparation

steps were described in the **section 11** of supporting information. **c** Self-healing mechanism (above) and the digital photos of a self-healing $EG_{50.3}$ sample. For self-healing, the sample was first scratched using a sharp blade, followed by soaking in $EG$ for 10 min and then UV irradiation at damage region (UV light intensity: $10 \, mW \cdot cm^2$, 2 min). The sample was further annealed at 70°C for 2 h before measurements. **d** Strain-stress curves of scratch samples before and after self-healing. Inserted (top) are digital photos of self-healed splines and stretched self-repairing spline. The black lines highlight the healed region.

15 mL ammonia in a 250 mL flask containing was heated to 60 °C under a stirring condition, followed by adding 6 mL TEOS. The mixture was stirred at 60 °C for 8 h. Excess ammonia and unreacted reagents in the matrix were removed by washing with ethanol and the product was collected by high-speed centrifugation (a rate of 9000 r·min⁻¹ for 5 min.). The total washing times should not be less than 6.

### SiO₂/PEG precursors

Take $EG_{40}$ precursor as an example. Briefly, 140 μL PEGDA containing photoinitiator (1 wt%) was added into $SiO_2$ ethanol suspensions. The solution was treated with vortex mixing and sonication, followed by annealing in an oven at 90 °C for 2 h to allow for solvent evaporation. When the ethanol solvent was evaporated, a supersaturated acrylate solution of $SiO_2$ was obtained. The precursor solution was nearly transparent and kept in dark for 5 h before use. The same protocol was used to prepare the other precursor solutions with different ratios of $SiO_2$ nanoparticles and PEGDA[36].

### Fabrication of photonic composite films

For example, 50 μL precursor solution was sandwiched between a glass slide (18 × 18 mm) and a hydrophobic cover silicon wafer (18 × 18 mm) (Supplementary Fig. 5). The sample was irradiated by UV light (365 nm, 10 mW·cm⁻²) for 2 min. After illumination, a vivid film formed and it was peeled off from the substrate for further study. This method could be applied to prepare large-scale films and the thickness of the films was controlled by the amount of the precursor solution and the area of the substrate.

### Light-induced growth

For growth, the swollen samples were subjected to UV light (365 nm, 10 mW·cm⁻²) for illumination with or without photomasks. The samples were then annealed at 70°C for 5 h in an oven.

### Reporting summary

Further information on research design is available in the Nature Portfolio Reporting Summary linked to this article.

## Data availability

The authors declare that the main data supporting the findings of this study are available within the article and its Supplementary Information files. Extra data are available from the corresponding author upon request. The source data underlying Figs. 2a, b, e-g, 3a, b, f, 4a and d and Supplementary Figs. 4, 6, 7, 10-13, 15, 16, 19 and 20 are provided as a Source Data file. Source data are provided with this paper.

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

## Acknowledgements

This work was supported by the National Natural Science Foundation of China (51973023 to J.C. and 52003035 to L.Y.), Sichuan Science and Technology Program (2021JDRC0014 to J.C. and 2021JDRC0106 to L.Y.) and China Postdoctoral Science Foundation (W03019023601004133 to J.X.). We also thank Ms. Songzi Xu and Junchang Guo at University of Electronic Science and Technology of China for their help on experiments and discussion.

## Author contributions

J. C. and J. X. conceived the concept. J.C. supervised the project. J.C. and J.X. designed the experiment. J.X., X.Y., C.Z., and Y.F. conducted the experiments. J.X., J.C., and L.L. wrote the manuscript. L.X., Y.L., L.Y., and S.D. discussed the results and commented on the manuscript at all stages. All authors contributed to the analysis and discussion of the data.

## Competing interests

The authors declare no competing interests.
