## [Peer Review File · Nature Communications]

Self-growing Photonic Composites with Programmable Colors and Mechanical PropertiesREVIEWER COMMENTS

Reviewer #1 (Remarks to the Author):

In the manuscript entitled “Self growing Photonic Composites with Programmable Colors and Mechanical Properties”, the authors proposed a swelling and UV curing method to regulate the color, mechanical properties of a SiO₂-polymer composite. The paper is well written, and the presented results are systematic. However, the concept of the dynamic swollen and the function of the transesterification catalyst have been already reported by the authors (Nature Communications 11, 963 (2020)), so the significance of this paper is in demonstrating a specific application of this mechanism in color patterning. This published paper should be discussed in the introduction as well as the literature regarding swelling and polymerization induced color patterning. The novelty should be emphasized in view of their previous publication in Nature Commun.

Besides, the following comments should be addressed before publication:

1. Will polymerization induce color change? In Figure 2a, color change seems not as obvious as in Figure 4a. Will the UV intensity influence the displayed color?
2. Most of the samples are prepared from swelling with nutrient along with polymerization. The sample prepared from one-step, for example, EG40-EG10.6, with the same chemical composition prepared with the same one-step polymerization as EG40, should be compared to illustrate the advantage of swelling.
3. The process of the “Sichuan facebook” is too simple to understand. As the swelling will lead to color change, why the white region remained white after multi-step swelling?
4. Will the thickness of the initial composite film influence the swelling process and the color regulation?

Reviewer #2 (Remarks to the Author):

The authors report on a strategy to fabricate photonic materials with controllable and patternable coloration based on a bio-inspired growth phenomenon. To this end, the authors use silica particles embedded in a polymer network and cleverly apply polymer chemistry to swell and rearrange the polymer network forming the matrix. As a result, the matrix grows, thus changing the interparticle distance of the silica particle at will. This, in turn, results in a macroscopic color shift. The authors elucidate the formation mechanism and color shift in detail, and explore other appealing features, such as self-healing capacities, control of mechanical properties, and, importantly, localized growths resulting in distinct color patterns. Overall, the manuscript is well written and provides an innovative novel approach to create well-defined photonic materials with attractive properties. I believe that the manuscript will be appealing to the broad readership of Nature Communications and recommend publication after the following questions have been clarified.

Photonic properties: On Page 7, the authors write that “in the ordered domains, SiO₂ nanoparticles arranged into ordered structures with uniform photonic bandgaps. Due to such a short-range ordered nature, the obtained color is angle-independent”. In the following, the Bragg equation based on ordered lattices is used to describe the bandgap and thus the color. This section is a bit confusing: Ordered materials are described by the Bragg equation for reflection at crystal lattices, but typically show angle-dependent color (the equation shown in the manuscript is for normal incidence). In contrast, regions with short-range order do not have an ordered lattice, but do show angle-independent color. I suggest to re-think which domains are relevant for the macroscopic color and rephrase accordingly. From the reflection data, SEM images, and the argumentation throughout the manuscript, a dominating effect of the crystalline regions seems likely, but to me, it remains unclear why the resultant color is angle-independent.

Transesterification and involved chemistry: In Figure 2, it may be worthwhile to include the chemistry of transesterification (i.e. the release of the initial network) as structural formulas, to combine the chemical (molecular) nature of the mechanism with the resultant changes in physical properties. If this will look too crowded, I suggest to at least show it as an SI figure. Figure S10 shows the reaction on the particle surface, but I feel that a complete presentation of the involved chemistries would be useful. In particular, what is not clear from the current description is how the network itself is reworked. Figure 1 suggests that crosslinking points also relax, which, I presume, is also due to transesterification of the acrylate units? The author discuss a similar effect in the framework of softening via alcohols. If this is correct, it should be discussed in a bit more detail in the main text and all relevant structural rearrangements shown via structural formulas, at least in the SI.

Refractive index contrast: In the description of Figure 2, the authors mention that “the monomer showed a similar refractive index with that of SiO₂”. In the SI, the authors also mention that the RIs between SiOs and matrix is very similar. If this is correct, where does the photonic bandgap actually come from? In an index-matched environment, one would not observe a photonic reflection at all.

Reviewer #3 (Remarks to the Author):

1. In Figure 1, a excellent cycle growth strategy of photonic crystal composite system is proposed. Does this growth strategy have limits? Or can it be repeated indefinitely? I hope the author can give more data and theoretical analysis on this problem.
2. In Figure 2e, with the growth of the size of the photonic crystal composite system, the spectral characteristic peak has a red shift. But at the same time, the half peak width has correspondingly increased, which means that the purity of the color is gradually decreasing. What is the reason for this phenomenon? I hope to give a detailed explanation.
3. As readers, the concepts of “transesterification reactions”(line192) and “transesterification catalyst”(line 196) still can not be well understood after reading the full text. It is hoped that the author will explain the mechanism of cyclic growth in more detail, which will help readers better understand this study.
4. In the methods (line 400), there is no description of the growth process of the photonic crystal composite system, so it is recommended to complete it.

Response to Reviewers (NCOMMS-22-08944)**(Original comments are in frames)****Response to Reviewer #1:**

In the manuscript entitled “Self growing Photonic Composites with Programmable Colors and Mechanical Properties”, the authors proposed a swelling and UV curing method to regulate the color, mechanical properties of a SiO₂-polymer composite. The paper is well written, and the presented results are systematic. However, the concept of the dynamic swollen and the function of the transesterification catalyst have been already reported by the authors (Nature Communications 11, 963 (2020)), so the significance of this paper is in demonstrating a specific application of this mechanism in color patterning. This published paper should be discussed in the introduction as well as the literature regarding swelling and polymerization induced color patterning. The novelty should be emphasized in view of their previous publication in Nature Commun.

Response: Thank you for your positive and insightful comments! We have discussed our previous work as well as the literature regarding swelling- and polymerization-induced color patterning in the introduction. The novelty is also emphasized. Specific discussions have been added to the introduction section in the revised manuscript and are marked in blue.

Page 3, paragraph 1:

In self-assembly methods, building blocks with uniform sizes are required to achieve ordered nanostructures, which leads to single or pseudo photonic bandgap and hence monochrome or iridescence. For many applications, multicolor patterns are necessary, such as cosmetics, apparel, security, bioanalysis, sensors, electrical appliances, and automobiles¹⁶. Patterning structural-color materials can significantly enhance their added value. To achieve multicolored images, several methods have been developed recently, including confinement deposition/swelling¹⁷, regioselective removal¹⁸, post-modification of inverse opal¹⁹, etc. Among them, selective swelling^{20,21} can even bring about interesting stimuli-responsiveness and reversibility. For swelling-induced patterning, region-dependent swellability is required, and the liquid-containing nature might also induce instability. By contrast, photopolymerization allows direct patterning by immobilizing compounds to modulate the bandgap²², but the variation in bandgap is limited due to the restriction of curing rigid substrates. Generally speaking, these methods typically involve elaborate fabrication techniques requiring superb synthetic skills, yet the obtained images are either ephemeral, low-resolution, or relatively dull, stiff, and low-controllable. Moreover, artificial structural color materials are often fragile and not reprocessable^{23,24}, in stark contrast to their glorious, durable, and lightweight natural counterparts. Therefore, it is highly desirable to develop facile approaches for fabricating bright, robust, and adaptive structure color patterns.

Page 5, paragraph 2:

Mimicking the growth of living organisms with synthetic building blocks is an emerging strategy for designing intelligent materials^{30,31}. We have recently developed a novel approach to enabling crosslinked polymers to incorporate externally-provided compounds to grow³². The growth

involves swelling and polymerization of precursors as well as homogenization of the newborn and original networks. Such design allows us to create structural surfaces from smooth substrates³³, endow rigid substrates with self-healing ability³⁴, and fabricate conductive hydrogel circuits³⁵. However, the growing systems currently available all possess simple polymer matrices, far away from the fine, ordered, complex composite structures observed in peacock's tail feathers. Herein, we propose to apply the growth concept to fabricate photonic composites with controlled structural coloration, patterning capability, and versatile mechanical properties.

The catalyst in the nutrient solution is thus designed to remove such restriction by inducing chain exchange between new and old networks to homogenize the matrices, allowing the system to further grow (see **Figure S2** for the chemical reactions designed for the growth). With this design, we can continuously change the photonic bandgap and system compositions for modulating materials' colors and mechanical properties on demand. Due to the spatiotemporal advantages of light, it is facile to generate multicolor images/patterns.

(1) Will polymerization induce color change? In Figure 2a, color change seems not as obvious as in Figure 4a. Will the UV intensity influence the displayed color?

Response 1: Thank you for your insightful comments! The films maintain their colors when they are irradiated homogeneously in the polymerization step. In our systems, polymerization might induce two possible consequences of making color change, i.e., polymerization-induced shrinkage and polymerization-induced mass transport. Since the swelling ratio is small (<9%), monomer-to-polymer conversion leads to negligible influence on material density. Therefore, no shrinkage (then change in lattice constant) occurs. On the other hand, homogeneous irradiation would induce in situ incorporation of the absorbed nutrients (monomer and crosslinker); thus, no mass transport is expected. In contrast, selective irradiation will induce remarkable color changes. The color in the irradiated region is red-shift while that in the unirradiated part is blue-shift, due to the polymerization-induced transport of the nutrients from the unirradiated to the irradiated regions (as evidenced in **Figures 4a & b**, and see more detailed discussion in the **Response 3**).

The images in **Figures 2a** and **4a** have different scale bars (1 cm for **Figure 2a** and 1 μm for **Figure 4a**), which makes them look different in color change. The image in **Figure 4a** is significantly enlarged and the color might be distorted. Sorry for missing the scale bars in the **Figure 2a**.

Three UV intensities (5, 10, and 20 $\text{mW} \cdot \text{cm}^{-2}$) have been applied to prepare the samples. UV-Visible reflection spectroscopy was further used to detect their optical properties. The obtained samples display similar colors (**Figure R1-1, Supplementary Figure 12 in the revised manuscript**). Therefore, we believe that the intensity of UV light would not affect the exhibited color once the light is strong enough to induce homogeneous polymerization.

Figure R1-1. UV-Visible spectra of the samples $EG_{40} - EG_{10.6}$ obtained from different UV light intensities (5, 10, and 20 $\text{mW} \cdot \text{cm}^{-2}$).

We have updated this information in the revised manuscript.

Page 12, Paragraph 1:

By contrast, the samples that were subsequently irradiated by UV light (365 nm, $10 \text{ mW} \cdot \text{cm}^{-2}$) to trigger photopolymerization preserved their after-swelling weights and colors, as well as became tougher (**Figure 2a & Supplementary Figure 11**). Three kinds of UV light with different intensities (5, 10, and $20 \text{ mW} \cdot \text{cm}^{-2}$) have been applied to induce the polymerization. The obtained samples show nearly identical spectra (**Supplementary Figure 12**), suggesting that the UV light intensities had a negligible impact on the displayed colors once the adopted light was strong enough to induce homogeneous polymerization.

(2) Most of the samples are prepared from swelling with nutrient along with polymerization. The sample prepared from one-step, for example, $EG_{40} - EG_{10.6}$, with the same chemical composition prepared with the same one-step polymerization as EG_{40} , should be compared to illustrate the advantage of swelling.

Response 2: Thanks for your valuable suggestions! We have prepared a sample of EG_{46} by one-step polymerization as a control to show the advantage of our growing strategy. The EG_{46} has the same compositions with $EG_{40} - EG_{10.6}$ $[(40+10.6)/(100+10.6)=45.8\%]$. It displays similar reflection peak positions as $EG_{40} - EG_{10.6}$ but its intensity is lower. Moreover, the half-peak width is broadened (**Figure R1-2, Supplementary Figure 14 in the revised manuscript**). These results indicate that the distribution of the colloidal bandgap is higher in the grown samples when the polymer contents are similar. Note that the ordered particle arrangement forms through a self-assembling process in the initiated samples (the seed for growth or the as-prepared sample). Increasing polymer fraction in this step will disturb the self-assembling process and thus decrease the ordered degree. In contrast, samples with well particle arrangement can be selected as seeds for growth. In the growth, the ordered arrangement is maintained, leading to narrow distribution of colloidal bandgap even in high polymer fractions. Besides this kind of advantage, the growth

strategy allows **spatial-selective change in bandgap for multicolor patterning**, which is exceptionally challenging for one-step polymerization. On the other hand, compared with the method of color regulation by swelling, the structural color of the grown sample can remain stable.

Figure R1-2. UV-Visible spectra of the samples EG_{46} and $EG_{40} - EG_{10.6}$. EG_{46} was obtained by one-step polymerization.

Page 13

“In comparison to the grown samples, the samples with similar composition could also be directly prepared via one-step polymerization. However, we found that the grown samples displayed sharper reflection peak than those samples obtained from one-step polymerization (Supplementary Figure 14), implying the advantage of our growth strategy in tuning the colors.”

(3) The process of the “Sichuan facebook” is too simple to understand. As the swelling will lead to color change, why the white region remained white after multi-step swelling?

Response 3: We are sorry for inducing such confusion. We used EG_{48} with single blue color as the starting substrate to prepare the “Sichuan facebook”. The sample was immersed in EG for swelling, followed by UV irradiation through photomasks. The grown sample was washed to remove the unreacted compounds for taking the picture (therefore, the light blue region remained its color). In every growth cycle, we changed the photomasks to control the growth regions. Here, we intend to show the constant color without irradiation using washing treatments. It is not always necessary to wash the unreacted nutrient to make multicolor patterns since the nutrients can be used directly for further growth. We have proved that direct swelling of the irradiated samples in nutrient solution for further irradiation can also lead to similar multicolor patterns. To prevent any misleading, we have added a schematic (Figure R1-3, Supplementary Scheme 2 in the revised supporting information) to show the general idea of making multicolor pattern and also detailed statement (section 11 in SI, Supplementary Figure 20 in the revised supporting information) to show the fabrication process of “Sichuan facebook” in the revised manuscript.

Figure R1-3. A schematic image to describe the procedure of preparing multicolor patterns.

11. Fabrication of "Sichuan facebook"

Similarly, we have fabricated the "Sichuan facebook" through our growing strategy. Three photomasks with different facebook shapes (glasses deposited by chromium, **Figure S20**) were employed to control the growing regions. EG_{48} with single blue color was used as the starting substrate, which was immersed in the nutrient EG for swelling (the sample became olive). The swollen sample was placed on a glass substrate and covered by a photomask, followed by UV light irradiation (intensity: $10 \text{ mW} \cdot \text{cm}^{-2}$) for 2 mins. During irradiation, the irradiated regions became yellow. The whole sample (still covered by the photomask) was then annealed in an oven at $70 \text{ }^\circ\text{C}$ for 5 h (no color change was observed during annealing). After annealing, the photomask was removed and the film was peeled off from the glass substrate. We found that the unirradiated regions were green. When we used ethanol/ CHCl_3 solution to wash the free nutrients, the irradiated regions were still yellow but the unirradiated regions turned back to blue. This growing cycle was repeated twice again by using different photomasks, finally leading to the multicolor "Sichuan facebook" pattern.

Figure R1-4. Three kinds of photomasks used in the preparation of "Sichuan facebook" in sequence.

Page 19, Paragraph 1:

Another significant advantage of this photonic composite system is its spatially selective growth capability by using patterned photoirradiation (**Figure 4a**, see **Supplementary Scheme 2** for the detailed patterning procedure of selective growth).

Page 19, Paragraph 1:

Localized growth of photonic crystal films indicated a novel approach for chromatic patterning. We fabricated a “Sichuan facebook” using a EG_{48} substrate with single blue color to demonstrate the flexibility of this approach (**Figure 4b**). The blue substrate became olivine after swelling in EG. When such swollen substrate was subjected to UV light under a mask, yellow “Sichuan facebook” profile was outlined. After the unreactive nutrients were removed, the irradiated regions maintained their yellow color while the unirradiated regions turned back to light blue. The substrate was then re-swollen with EG for further selective irradiations twice through different masks (**Supplementary Figure 20**), leading to the formation of new structural colors and a clear multicolor image.

(4) Will the thickness of the initial composite film influence the swelling process and the color regulation?

Response 4: In our method, the thickness of the initial composite film is determined by the volume of the $SiO_2/PEGDA$ precursor. We have prepared three samples with average thicknesses ranging from 0.1 to 0.2 mm by increasing the volume of the precursor solution from 20 to 70 μL , and studied their swelling processes (**Figure R1-5, Supplementary Figure 5 in the revised manuscript**). The samples could swell the tested nutrient solutions with equilibrium swelling ratios of 6.3 wt% (0.1 mm), 6.3 wt% (0.15 mm), and 6.1 wt% (0.2 mm), respectively. These results indicated that the thickness of the initial sample within 0.1–0.2 mm has bare effects on the swelling behavior. Subsequently, these three initial samples were soaked in nutrients EG for one night, followed by UV light irradiation (intensity: 10 mW cm^{-2}) for 2 minutes. The obtained samples are then annealed at 70°C for 5 h in an oven. Finally, the above samples are cleaned by immersing them in ethanol/ $CHCl_3$ solution for 3 h. Similarly, the optical properties of grown samples were characterized by using UV-visible reflectance spectroscopy. These films with different thicknesses show nearly the same spectra (**Figure R1-6, Supplementary Figure 6 in the revised manuscript**). Therefore, it can be concluded that the thickness of the initial sample rarely influences color regulation within a range from 0.1 to 0.2 mm.

Here the thickness range was selected based on the following consideration. The initial composite film should not be too thin to get self-standing and macroscopically uniform films (for getting integrated films from peeling off and evaluating their mechanical properties). On the other hand, the initiated films should not be too thick to prevent defect formation. The defects formed in thick samples may weaken the structural color intensity (Adv. Funct. Mater. **2014**, *24*, 817-825). Moreover, most of the incident light interacts with only about a dozen layers near the surface

(ChemPhysChem **2010**, *11*, 579-583). Therefore, changing the thickness in a wide range is not necessary.

Figure R1-5. Swelling curves of EG_{40} with different thicknesses (0.10, 0.15 and 0.20 mm) in EG nutrient.

Figure R1-6. UV-Visible spectra of grown specimens made from EG_{40} seeds with thicknesses of 0.10, 0.15 and 0.20 mm.

Moreover, we have added these additional results in the revised manuscript and supporting information.

Page 7, Line 10:

The formation mechanism and driving force of SiO_2 forming an ordered structure were polymerization-induced colloidal assembly which had been carried out by Ge et al³¹. The average thickness of these vivid films is 0.14 mm. We found that changing film thickness (in the range of 0.1 - 0.2 mm) made no contribution to film swellability and color (Supplementary Figure 5 & 6). Thus, the films with a thickness of 0.14 mm were employed for further study.

Response to Reviewer #2:

The authors report on a strategy to fabricate photonic materials with controllable and patternable coloration based on a bio-inspired growth phenomenon. To this end, the authors use silica particles embedded in a polymer network and cleverly apply polymer chemistry to swell and rearrange the polymer network forming the matrix. As a result, the matrix grows, thus changing the interparticle distance of the silica particle at will. This, in turn, results in a macroscopic color shift. The authors elucidate the formation mechanism and color shift in detail, and explore other appealing features, such as self-healing capacities, control of mechanical properties, and, importantly, localized growths resulting in distinct color patterns. Overall, the manuscript is well written and provides an innovative novel approach to create well-defined photonic materials with attractive properties. I believe that the manuscript will be appealing to the broad readership of Nature Communications and recommend publication after the following questions have been clarified.

Response: Thank you for recommending the publication of our manuscript! We have addressed your concerns with additional experiments.

(1) Photonic properties: On Page 7, the authors write that “in the ordered domains, SiO₂ nanoparticles arranged into ordered structures with uniform photonic bandgaps. Due to such a short-range ordered nature, the obtained color is angle-independent”. In the following, the Bragg equation based on ordered lattices is used to describe the bandgap and thus the color. This section is a bit confusing: Ordered materials are described by the Bragg equation for reflection at crystal lattices, but typically show angle-dependent color (the equation shown in the manuscript is for normal incidence). In contrast, regions with short-range order do not have an ordered lattice, but do show angle-independent color. I suggest to re-think which domains are relevant for the macroscopic color and rephrase accordingly. From the reflection data, SEM images, and the argumentation throughout the manuscript, a dominating effect of the crystalline regions seems likely, but to me, it remains unclear why the resultant color is angle-independent.

Response 1: Thank you for your insightful comments! We agree that it is important to explain the mechanism of the angle-independent color. To this end, we have conducted thorough literature research and re-checked our data. We prepared the initial samples by a reported method (Adv. Funct. Mater. **2014**, *24*, 817-825). The obtained samples show angle-independent colors. It is consistent with the reported results. This angle-independent feature indicates the formation of amorphous ordered arrays rather than colloidal crystal (*J. Mater. Chem.*, **2012**, *22*, 23299). Large-scale SEM image shows that the grown specimens consist of both long-range-ordered and short-range ordered domains (**Figure R2-1, Supplementary Figure 7 in the revised manuscript**). The long-range ordered domains were minorities. They were dispersed randomly in short-range ordered domains, forming the amorphous structures. As a result, the colors are angle-independent

(**Supplementary Figure 8**). “The short-range order also makes the colloidal materials show structural colors through constructive interference. As the scattered lights from colloidal particles in amorphous arrays are not perfectly in-phase, the interference occurs at a relatively wide range of wavelength, resulting in weak reflection at a broad range of the wavelength. The central wavelength of the reflection is approximately located in $2dn_{eff}$, where d is average interparticle distance” (ACS Appl. Mater. Interfaces 2019, 11, 14485). Therefore, we believe it is still reasonable to use the equation to evaluate the reflection peaks.

To avoid any misunderstanding, we have modified the SEM images and discussed this in more detail in the revised manuscript and supporting information.

Figure R2-1. SEM images of grown sample $EG_{40} - EG_{10.6}$ synthesized in our work.

Page 8, The first line last,

Scanning electron microscopy (SEM) investigation illustrates that the samples consisted of crystal-like and short-range ordered domains (**Supplementary Figure 7**). The long-range ordered structures would lead to iridescence while the short-range ordered one would result in angle-independent colors.³⁷ In our samples, long-range ordered domains were minorities. They were dispersed randomly in short-range ordered matrices, forming amorphous structures. As a result, the colors are angle-independent (**Supplementary Figure 8**).

Page 9, paragraph 1,

where d refers to the average interparticle distance, n_{eff} indicates the mean refractive index of the composites, and m is the order of reflection ($m = 1, 2, \dots$).

(2) Transesterification and involved chemistry: In Figure 2, it may be worthwhile to include the chemistry of transesterification (i.e. the release of the initial network) as structural formulas, to combine the chemical (molecular) nature of the mechanism with the resultant changes in physical properties. If this will look too crowded, I suggest to at least show it as an SI figure. Figure S10 shows the reaction on the particle surface, but I feel that a complete presentation of the involved chemistries would be useful. In particular, what is not clear from the current description is how the network itself is reworked. Figure 1 suggests that

crosslinking points also relax, which, I presume, is also due to transesterification of the acrylate units? The author discuss a similar effect in the framework of softening via alcohols. If this is correct, it should be discussed in a bit more detail in the main text and all relevant structural rearrangements shown via structural formulas, at least in the SI.

Response 2: Thanks for your valuable comments! We agree that a clear description of transesterification would make the manuscript more readable. The specific chemical reaction equations have been added to the revised supporting information. Here, we illustrate the chemical processes of initial network release using the initial sample EG_{40} grown in nutrient solution B as an example (**Figure R2-2, Supplementary Figure 2 in the revised manuscript**).

Chemical reaction equations

1. Polymerization:

2. Transesterification:

Figure R2-2. 1. Initial (red one) and newborn (green one) polymer networks formed via photopolymerization. 2. The grown matrices are homogenized via transesterification reactions between the original and newborn polymer networks.

Moreover, we have added these contents in the revised manuscript.

Page7, line 7:

The catalyst in the nutrient solution is thus designed to remove such restriction by inducing chain exchange between new and old networks to homogenize the matrices, allowing the system to grow further (see Supplementary Figure 2 for the chemical reactions designed for the growth).

Supporting information:

Here, we illustrate the chemical processes of initial network release using the initial sample EG_{40} grown in nutrient solution B as an example.

(3) Refractive index contrast: In the description of Figure 2, the authors mention that “the monomer showed a similar refractive index with that of SiO_2 ”. In the SI, the authors also mention that the RIs between SiO_2 and matrix is very similar. If this is correct, where does the photonic bandgap actually come from? In an index-matched environment, one would not observe a photonic reflection at all.

Response 3: We agree that no photonic reflection should be observed in an index-matched environment. In our system, the refractive index (RI) of particles (1.45) is quite close to that of polymer matrices (1.47). Therefore, the variation in polymer fraction hardly change the average RI of the system. The low RI contrasts also lead to relatively weak reflectance of colloidal films (note: RI contrast is low but still exists). However, besides RI contrast, the reflectance also depends on the film thickness. Increasing film thickness will enhance the reflection effect by causing strong multiple scattering (Chem. Mater. **2021**, 33, 1714). In the manuscript, we choose EG_{40} with an average thickness of 0.14 mm as an initial sample for our study. By using the growing strategy, the reflectance intensities of the obtained growth sample films are above 45%, which is sufficient for the human eye to observe the presence of structural color.

We have discussed this in the revised manuscript.

Page14, paragraph 1:

Furthermore, in our system, the polymer showed a similar refractive index n with that of SiO_2 (SiO_2 : 1.45; polymer matrix: 1.47 at 20°C), and therefore, the n_{eff} could be considered as a constant during growth. Note that the small refractive index contrast would lead to weak reflection. In our case, the samples were quick thick such that strong multiple scattering would occur to induce brilliant structural colors³⁸.

Response to Reviewer #3:

(1) In Figure 1, a excellent cycle growth strategy of photonic crystal composite system is proposed. Does this growth strategy have limits? Or can it be repeated indefinitely? I hope the author can give more data and theoretical analysis on this problem.

Response 1: Thank you for your insightful comments! In our design, the growth involves three steps. At first, the crosslinked composites are swollen by a nutrient solution consisting of monomer, crosslinker, photoinitiator, and catalyst. The swollen samples are irradiated to induce in-situ polymerization to form double-network structures in which the polymer chains in the original network adopt extending conformations. Finally, the samples are annealed to trigger transesterification-based chain exchange reactions. This homogenization can relax the tension generated in the extending chains, resulting flexible conformations again. As a result, the grown samples can swell nutrients for next growing cycle again. We found that the grown samples display near the same swelling ratio as the seeds. Therefore, we believe that the growing cycle can be repeated indefinitely in principle. Moreover, the samples could maintain their ordered structures showing structural colors. As shown in the **Figure 2e**, the seed EG_{40} had underwent in sequence five growing cycles to red-shift the reflection peak (λ) from 481 to 690 nm, indicating that the growth-induced expansion of the lattice parameters allows the system to vary its photonic bandgaps and thus color continuously in a wide range. Actually, for structural color applications, it is not necessary to repeat the growth indefinitely since human eye can only detect the light in the visible range of 380-800 nm.

The growing cycle has been clearly described in Paragraph 3 (see below). We have also modified our statement to further emphasize this in the revised manuscript.

“the growth is achieved via homogeneously swelling the crosslinked polymer matrices using a nutrient solution consisting of monomer, crosslinker, photoinitiator, and catalyst simultaneously, followed by light-induced polymerization. A new-old double network structure forms, in which the original network is stretched. Note that the stretched conformation would restrict the uptake of more nutrient solution for further growth. The catalyst in the nutrient solution is thus designed to remove such restriction by inducing chain exchange between new and old networks to homogenize the matrices, allowing the system to further grow”.

Page 12

“As a result of such relaxation, the grown samples could swell nutrients again, showing similar swelling ratios with the original samples (**Figure 2b**). This feature allowed repeated growth. To illustrate the role of this homogeneous step in growth, a control sample without transesterification catalyst was treated under the same process aforementioned. As expected, this catalyst-free sample was nearly non-swellable to the nutrient solution after the first growing cycle. In principle, the growing cycles could be repeated indefinitely to continuously red-shift the color, even though we only focused on the visible range in the current study.

(2) In Figure 2e, with the growth of the size of the photonic crystal composite system, the spectral characteristic peak has a red shift. But at the same time, the half peak width has correspondingly increased, which means that the purity of the color is gradually decreasing. What is the reason for this phenomenon? I hope to give a detailed explanation.

Response 2: Thanks for your insightful comments! Our initial photonic composite films (seeds) are prepared by precipitating colloidal crystal clusters in a supersaturated solution of SiO₂ nanoparticles, followed by photopolymerization to get amorphous structures (Adv. Funct. Mater. **2014**, *24*, 817-825). The samples consisted of both long-range ordered and short-range ordered domains (**Supplementary Figure 7**). Since the samples show angle-independent colors, the short-range ordered domains determine the color. The reflection peaks in **Figure 2e** should be assigned to the average lattice constant of the short-range ordered structures. With increasing growth cycles, the full-width-at-half-maxima (FWHM) of the reflection peak increase slightly, indicating a decrease in the ordered degree of these structures. We attribute this decrease to the inhomogeneous nature of the short-range ordered structures. The regions containing more polymer matrices are softer and would grow faster than other regions. With increasing the growth cycles, such inhomogeneity is also magnified, leading broader distribution of the lattice constant. In despite of this decrease, the high-polymer-content samples obtained by growth still display narrower FWHM compared to the samples directly prepared via one-step polymerization (**Figure R3-1, Supplementary Figure 14 in the revised manuscript**).

Figure R3-1. UV-Visible spectra of the samples EG_{46} and $EG_{40} - EG_{10.6} \cdot EG_{46}$ was obtained by one-step polymerization. The EG_{46} has the same compositions with $EG_{40} - EG_{10.6}$ [(40+10.6)/(100+10.6)=45.8%]

Page 12, Line 10:

With the increase in polymer matrices (then lattice distances), the reflection peak (λ) shifts from 481 to 690 nm. These reflection peaks were surprisingly intensive and narrow, implying a highly ordered arrangement of SiO₂ colloid crystals. The reflection peaks became slightly broader with growth, implying a decrease in ordered degree of the SiO₂ colloid crystals. We attributed this decrease to the growth-induced magnification of the inhomogeneity of the short-range ordered structures that induced the structural colors. In comparison to the grown samples, the samples with similar composition could also be directly prepared via one-step polymerization. However, we found that the grown samples displayed sharper reflection peak than those samples obtained from one-step polymerization (**Supplementary Figure 14**), implying the advantage of our growth

strategy in tuning the colors.

(3) As readers, the concepts of “transesterification reactions” (line192) and “transesterification catalyst” (line 196) still can not be well understood after reading the full text. It is hoped that the author will explain the mechanism of cyclic growth in more detail, which will help readers better understand this study.

Response 3: Thanks for your insightful comment! The growth starts from crosslinked polymer composites (initial sample) and include following three steps. At first, the crosslinked composites are swollen by a nutrient solution consisting of monomer, crosslinker, photoinitiator, and catalyst. The swollen samples are irradiated to induce in-situ polymerization to form double-network structures in which the polymer chains in the original network adopt extending conformations. Finally, the samples are annealed to trigger transesterification-based chain exchange reactions. This homogenization can relax the tension generated in the extending chains, resulting flexible conformations again. As a result, the grown samples can swell nutrients for next growing cycle again. To better understand the concepts of “transesterification reactions”, we have added the chemical reaction equations of transesterification using the initial sample EG_{40} grown in nutrient solution *B* as an example in the revised manuscripts (**Figure R3-2, Supplementary Figure 2 in the revised manuscript**).

Chemical reaction equations

1. Polymerization:

2. Transesterification:

Figure R3-2. 1. Initial (red one) and newborn (green one) polymer networks formed via photopolymerization. 2. The grown matrices are homogenized via transesterification reactions between the original and newborn polymer networks.

We have added more discussion about this in the revised manuscript.

Page 4, Paragraph 3:

The catalyst in the nutrient solution is thus designed to remove such restriction by inducing chain exchange between new and old networks to homogenize the matrices, allowing the system to further grow (see Supplementary Figure 2 for the chemical reactions designed for the growth).

Supporting information:

Here, we illustrate the chemical processes of initial network release using the initial sample EG_{40} grown in nutrient solution B as an example.

(4) In the methods (line 400), there is no description of the growth process of the photonic crystal composite system, so it is recommended to complete it.

Response 4: Thanks for your reminding! We have described the growth process of the photonic crystal composite system in the section on **Methods**. The specific contents can be seen in the revised manuscript.

Page 25, paragraph 2:

Methods

Light-induced growth: For growth, the swollen samples were subjected to UV light (intensity: 10 mW • cm⁻²) for 2 minutes illumination. The samples were then annealed at 70°C for 5 h in an oven.

REVIEWERS' COMMENTS

Reviewer #2 (Remarks to the Author):

The authors have fully answered the reviewer's comments with abundant supporting description and data, I think the revised manuscript could be accepted for publication.

Reviewer #3 (Remarks to the Author):

The authors have addressed all concerns raised by myself (and the other reviewers) and I believe that the manuscript can be published. As a minor note, there is a typo in their added section on Page 14. "the samples were quick thick.." should be "quite".

Reviewer #4 (Remarks to the Author):

It has been modified and recommended to be published.

Point-by-Point Response to Reviewers (NCOMMS-22-08944A)

Reviewer #2 (Remarks to the Author):

The authors have fully answered the reviewer's comments with abundant supporting description and data, I think the revised manuscript could be accepted for publication.

Response: We thank very much for the reviewer's comments to the manuscript.

Reviewer #3 (Remarks to the Author):

The authors have addressed all concerns raised by myself (and the other reviewers) and I believe that the manuscript can be published. As a minor note, there is a typo in their added section on Page 14. "the samples were quick thick." should be "quite".

Response: We appreciate very much for the reviewer's suggestions. We have revised "quick" to "quite" in the revised manuscript.

Reviewer #4 (Remarks to the Author):

It has been modified and recommended to be published.

Response: Thank you for recommending the publication of our manuscript!